# Softening of a flat phonon mode in the kagome ScV$_6$Sn$_6$

A. Korshunov [1,11], H. Hu [2,11], D. Subires [2,11], Y. Jiang [3,4,11], D. Călugăru [5,11], X. Feng [2,6,11], A. Rajapitamahuni [7], C. Yi [6], S. Roychowdhury [6], M. G. Vergniory [2,6], J. Strempfer [8], C. Shekhar [6], E. Vescovo [7], D. Chernyshov [9], A. H. Said [8], A. Bosak [1], C. Felser [6], B. Andrei Bernevig [2,5,10] ✉ & S. Blanco-Canosa [2,10] ✉

Geometrically frustrated kagome lattices are raising as novel platforms to engineer correlated topological electron flat bands that are prominent to electronic instabilities. Here, we demonstrate a phonon softening at the $k_z = \pi$ plane in ScV$_6$Sn$_6$. The low energy longitudinal phonon collapses at ~98 K and $\mathbf{q} = \frac{1}{3}\frac{1}{3}\frac{1}{2}$ due to the electron-phonon interaction, without the emergence of long-range charge order which sets in at a different propagation vector $\mathbf{q}_{CDW} = \frac{1}{3}\frac{1}{3}\frac{1}{3}$. Theoretical calculations corroborate the experimental finding to indicate that the leading instability is located at $\frac{1}{3}\frac{1}{3}\frac{1}{2}$ of a rather flat mode. We relate the phonon renormalization to the orbital-resolved susceptibility of the trigonal Sn atoms and explain the approximately flat phonon dispersion. Our data report the first example of the collapse of a kagome bosonic mode and promote the 166 compounds of kagomes as primary candidates to explore correlated flat phonon-topological flat electron physics.

The search for quantum materials with novel forms of entangled fermion-fermion and fermion-boson interactions has led to the discovery of new emergent electronic phases of matter with charge order, epitomized in the family of high Tc superconductors[1,2] and in the recently discovered hexagonal kagome metals[3–5]. The kagome lattice, hosting large electron densities of states at the Fermi level, has recently emerged as a rich platform to study the interplay between topology and correlated physics[6]. Topological flat bands derived from the destructive interference of the electronic wavefunction, van Hove singularities (vHs) at the Brillouin zone boundary (M point), and Dirac crossings at the BZ corner (K point) lurk at, beneath, and above the Fermi level[7,8] depending on the electron number. $d$-electrons provide a path for interactions and the intertwining of different exotic orders. Among those, the interplay between multi-component hexagonal charge density waves (CDWs)[9], magnetism[10,11], nematic order[12]

and superconductivity are extensively studied as candidates to explore strongly correlated topological physics. While the band structure is very complicated, and cannot be purely understood by the usual kagome flat band argument[13], which would give an incorrect counting of the number of flat bands in these $d$-orbital systems hybridized with $p$ orbitals of close-by atoms[14], it is clear that both wavefunction topology and interactions are important for a wide range of these materials, such as FeGe and the 166 series MT$_6$Z$_6$ (M = metallic elements, such as Mg and rare-earth elements; T = transition metal, V, Cr, Mn, Fe, Co, Ni; Z = main group elements, Si, Ga, Ge, Sn)[15,16].

Magnetism aside, strong attention is being paid to the AV$_3$Sb$_5$ (A = K, Cs, and Rb) system, hosting electronic ordering competing with superconductivity at lower temperature[17,18]. Although recently questioned by Kerr measurements[19], the CDW in AV$_3$Sb$_5$ (hereafter AVS), defines an electronic chirality[20] that might additionally break the time-

[1]European Synchrotron Radiation Facility (ESRF), BP 220, F-38043 Grenoble, France. [2]Donostia International Physics Center (DIPC), Paseo Manuel de Lardizábal, 20018 San Sebastián, Spain. [3]Beijing National Laboratory for Condensed Matter Physics, and Institute of Physics, Chinese Academy of Sciences, Beijing 100190, China. [4]University of Chinese Academy of Sciences, Beijing 100049, China. [5]Department of Physics, Princeton University, Princeton, NJ 08544, USA. [6]Max Planck Institute for Chemical Physics of Solids, 01187 Dresden, Germany. [7]National Synchrotron Light Source II, Brookhaven National Laboratory, Upton, NY 11973, USA. [8]Advanced Photon Source, Argonne National Laboratory, Lemont, IL 60439, USA. [9]Swiss-Norwegian BeamLines at European Synchrotron Radiation Facility, Grenoble, France. [10]IKERBASQUE, Basque Foundation for Science, 48013 Bilbao, Spain. [11]These authors contributed equally: A. Korshunov, H. Hu, D. Subires, Y. Jiang, D. Călugăru, X. Feng. ✉e-mail: bernevig@princeton.edu; sblanco@dipc.org

reversal symmetry[21]. The AVS lattice dynamics across the multiple $\mathbf{q}_{CDW}$ is highly controversial, despite ab initio calculations which predict soft modes at M and L points of the BZ[22,23]. The softening of a particular phonon branch is an elegant hallmark of the CDW transition, as first incarnated by Peierls in the perfectly nested Fermi surface of a one-dimensional metal[24], and defines a soft lattice dynamics of a second-order phase transition[25]. Nevertheless, the nesting scenario fails in higher dimensions; the electronic susceptibility does not fully diverge, the Fermi surface is not perfectly nested, or the experimental $\mathbf{q}_{CDW}$ does not match the nesting wavevectors[26]. Initial arguments based on the scattering between saddle points in AVS were supported by the presence of nested components of the Fermi surface[27,28], but signatures of the fermion-boson many body interplay (Kohn anomaly) are not manifested experimentally[29].

Alternatively, a periodic lattice distortion can also be stabilized in the presence of a finite orbital and momentum-dependent electron-phonon interaction (EPI)[30], provided that the electronic degrees of freedom enhance the electron-phonon matrix elements. The concept of momentum-dependent EPI is strongly supported by the experimental data; a broad momentum spread of the phonon softening is observed by x-ray scattering[31–33], a significant gap ratio ($\Delta/k_B T_{CDW}$) exceeding the BCS theory[34] or the presence of a pseudogap at $T > T_{CDW}$[35,36]. Following this argument, the dynamics of the CDW in AVS seems to follow an order-disorder transformation type (strong coupling limit)[37,38] that leads to a first-order phase transition and the absence of the lattice collapse at the critical temperature, $T_{CDW}$. The magnetic kagome FeGe also features a first-order like CDW transition[4] without phonon softening[39], but strongly intertwined with its magnetic order[40].

Very recently, a first order-like CDW with propagation vector $\frac{1}{3}\frac{1}{3}\frac{1}{3}$ has been observed in the ScV$_6$Sn$_6$ (hereafter SVS) kagome lattice (166 family) at $T_{CDW} = 95$ K[41], with similar filling of the V $d$-orbitals as AVS (importantly, away from the flat bands). However, unlike in AVS, the Sn $p$-bands in SVS contribute weakly (only through the trigonal and not hexagonal Sn) to the Fermi surface, thus the two vHs at M are derived from the $d$-orbitals of the V-sublayer. Initial angle-resolved photoemission (ARPES) data[42], supported by optical spectroscopy[43], revealed no energy gap at the Fermi level, in contrast to the 20 meV CDW gap observed by scanning tunneling spectroscopy (STM)[44]. Moreover, the comparison of different members of the 166 kagome family (some at different electron numbers) with and without periodic modulations suggests an unconventional nature of the CDW, strongly coupled with its lattice dynamics[45]. This is particularly interesting, since the propagation vector of the charge modulations in SVS is not compatible with the scattering between high symmetry points of the BZ, marking a clear distinction with the primary order parameters of AVS and FeGe[46]. Furthermore, recent ab initio calculations show a rich landscape of lattice instabilities - yet physically not understood- where the Sc and Sn vibrations play a dominant role[47]. Nevertheless, despite intense efforts to microscopically understand the nature of the CDW phases in materials, a direct experimental and theoretical proof of the fermion-boson coupling in the kagome lattice is still missing.

Here, we use a combination of high-resolution elastic and inelastic x-ray (IXS) and diffuse scattering (DS), ARPES, density functional theory (DFT) and analytical effective models to experimentally reveal and theoretically understand a softening of the $k_z = \pi$ plane of the ScV$_6$Sn$_6$ kagome net. We observe that a phonon with propagation vector $\frac{1}{3}\frac{1}{3}\frac{1}{2}$ collapses at the same temperature, $T_{CDW} \sim 98$ K, as the long-range CDW order with propagation vector $\frac{1}{3}\frac{1}{3}\frac{1}{3}$ sets in. The softening of the low energy branch is broad in momentum space, unveiling a strong momentum dependence of the electron-phonon interaction (EPI) which only influences some phonons on certain parts of the Brillouin Zone (BZ). Hard x-ray diffraction and theoretical modeling reveal that the soft phonon-driven condensation consists of a longitudinal mode characterized by an out-of-plane vibration of trigonal Sn (hereafter

Sn$^T$) located in the trigonal environment. This mode starts flat at high temperatures and acquires (small) dispersion due to the electron renormalization, as we show experimentally and theoretically through full ab initio calculations and intuitive effective field theory models.

## Results and discussion

We start with a brief overview of the structure and electronic band dispersion of SVS. Unlike AVS, where the Sb atoms lie in the V plane, Sn$^T$ atoms sit above and below the kagome net (blue spheres in Fig. 1A and B). Although this displacement does not corrugate the kagome plane, it actually introduces considerable modifications in their band structure, which becomes even more complicated due to four possible surface terminations: two V$_3$Sn with different bucklings, ScSn$_2$, and honeycomb Sn in the middle, among which only two are reported experimentally[48] (Supplementary Note 11). The Fermi surface (FS) topology measured at 7 K with 90 eV photons that probe the $k_z = 0.4$ plane of the bulk BZ is displayed in Fig. 1D. The measured band dispersion is shown along the high-symmetry directions in Fig. 1E, F, with the ab initio bands included in Fig. 1G for comparison. The large hexagonal FS centered at the $\overline{\Gamma}$ point is surrounded by a large triangular and small circular Fermi pockets centered at the K point, derived from the $d$ orbital character of V atoms. In addition, the Fermi surface presents high spectral intensity at the M points of the BZ due to the proximity of the vHs at the Fermi level. In Fig. 1E, F, the band dispersion along the $\overline{M} - \overline{K} - \overline{\Gamma}$ and $\overline{\Gamma} - \overline{M}$ segments are displayed in two different surface terminations, namely, V$_3$Sn and ScSn$_2$, respectively. At the $\overline{\Gamma}$ point and E-E$_F$ = -0.8 eV, we have observed a mismatch of -80 meV, presumably due to surface effects of one of the terminations or to a small different doping level in some regions of the crystal during the growth process. Consequently, we have introduced an identical energy shift in the ab initio bands from Fig. 1G to qualitatively match the ARPES data and the DFT calculations, and in agreement with the recent ARPES results[44,45,49]. Along the $\overline{\Gamma} - \overline{M}$ line, the ab initio bands that cross the Fermi level in the pristine phase (black lines) are absent in the experiment. Additionally, the signal suppression at ~−0.2 eV from Fig. 1E, F indicated by the red arrow, matches the avoided band crossing obtained from DFT in the CDW phase. The apparent Dirac crossing at ~−0.8 eV near the $\overline{\Gamma}$ point seen in Fig. 1F stems from the symmetry-enforced suppression of the signal at the $\overline{\Gamma}$ point as a result of the interplay between the orbital character of the corresponding bands and the polarization of the incoming photons. Finally, the surface states of SVS, shown by the orange arrow in Fig. 1E, F and featuring prominent spectral weight in the vicinity of the Fermi level along the $\overline{M} - \overline{K}$ direction are consistent with the ab initio calculations from Fig. 1G (Supplementary Note 11).

From our hard x-ray resonant diffraction data, Fig. 1H (Supplementary Note 6), we find CDW peaks at h $\pm \frac{1}{3}$ k $\pm \frac{1}{3}$ l $\pm \frac{1}{3}$, in agreement with the neutron diffraction data[41]. The absorption spectrum at the V K-edge (Fig. 1H) exhibits a pre-edge feature (labeled as A), assigned to the dipole forbidden transition $1s \rightarrow 3d$ and a strong peak 15 eV above the pre-edge (B), assigned to the dipole-allowed transition $1s \rightarrow 4p$. Energy dependence of the CDW reflection shows a small hump at the V pre-edge, which arises due to a combination of strong $3d - 4p$ mixing and overlap of the metal 3d orbitals with the 5p orbitals of Sn[50]. The CDW reflections start at ~98 K, with its maximum intensity at 92 K, coinciding with $T_{CDW}$ observed in resistivity, and a maximum in-plane correlation length of 80 nm.

Next, we proceed with the analysis of the diffuse scattering (DS) maps, hunting for CDW precursors across the reciprocal space (RS). Figure 2A shows the reconstructed maps spanning the h h $\frac{13}{2}$ plane at 300 K. Clear diffuse signals with triangular shape are visible at h $\pm \frac{1}{3}$ h $\pm \frac{1}{3}$ l $= \frac{1}{2}$ positions (H point) and grow in intensity upon cooling down to 100 K, below which it rapidly vanishes. On the other hand, no diffuse intensity is present either with propagation vectors $\frac{1}{3}\frac{1}{3}\frac{1}{2}$, $\frac{1}{2}0\frac{1}{2}$ (L), 0 0 $\frac{1}{2}$ (A)[47], $\frac{1}{3}\frac{1}{3}$ 0 (K) and $\frac{1}{2}$ 0 0 (M) in the whole temperature range above

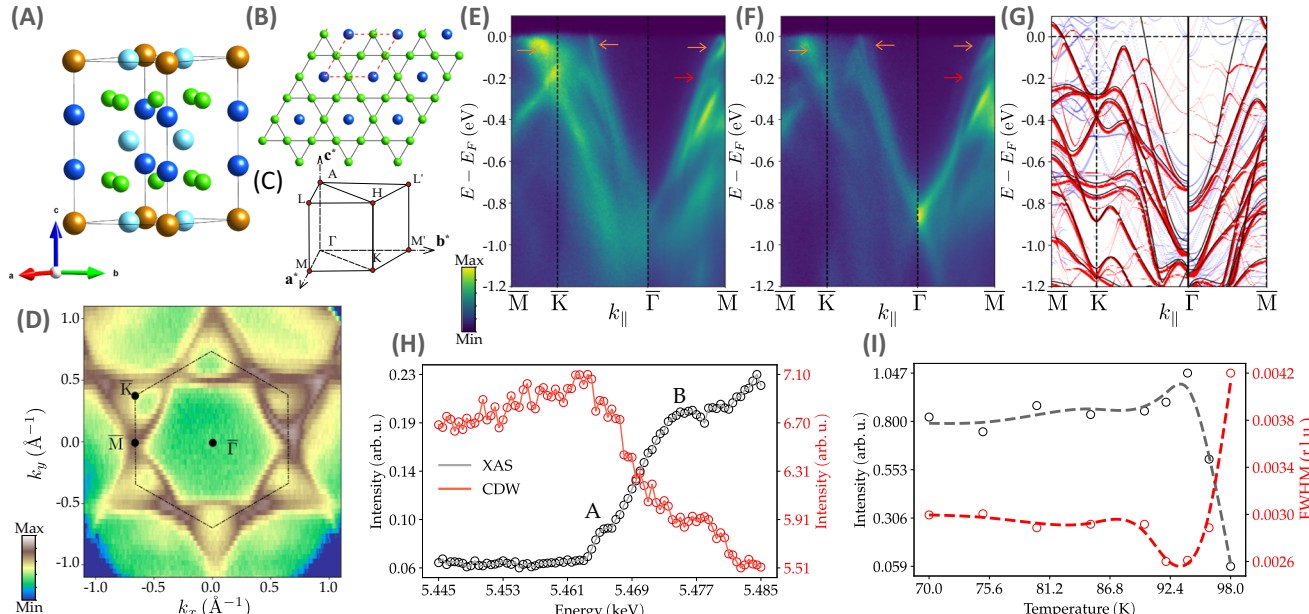

**Fig. 1 | Crystal and electronic band structure of ScV₆Sn₆. A** Side view of the ScV₆Sn₆ structure. Green and gold balls stand for V atoms that define the kagome net and Sc, respectively. Blue and cyan denote the trigonal Sn (Sn$^T$) and hexagonal Sn (Sn$^H$) atoms. Note that the Sn$^T$ atoms are lifted above and below the V kagome plane. **B** Top view of the kagome net, highlighting the V and Sn$^T$ atoms. Red dashed lines denote the unit cell. **C** Brillouin zone of the space group $P6/mmm$ (191) and the main symmetry directions. **D** Fermi surface map of ScV₆Sn₆ at 7 K taken with horizontal polarized light and $E_i$ = 90 eV ($k_z$ = 0.4 Å$^{-1}$). **E** Band dispersions along the high symmetry direction $\overline{M} - \overline{K} - \overline{\Gamma} - \overline{M}$ acquired at 7 K with horizontal (**H**) and (**F**) vertical (**V**) polarized light (Supplementary Note 11). The $\overline{\Gamma} - \overline{M}$ and $\overline{M} - \overline{K} - \overline{\Gamma}$ directions were taken with a ScSn₂ and with V₃Sn surface termination, respectively (see Supplementary Note 11 for further details). Orange and red arrows identify the surface states and the gap opening due to the avoided band crossing, respectively. **G** DFT calculations reproduce qualitatively the bulk and surface states of SVS, where black, red, and blue bands stand for the pristine bands, unfolded CDW bands, and surface bands, respectively. The bands along $\overline{M} - \overline{K} - \overline{\Gamma}$ are shifted upwards by ~80 meV to match the ARPES bands. **H** Resonant hard x-ray scattering showing the energy profile of the $\frac{1}{3}\frac{1}{3}\frac{10}{3}$ CDW peak at the V K-edge. See the main text for the explanation of the A and B peaks. **I** Temperature dependence of the intensity and linewidth of the CDW peak at the V K-edge (Supplementary Note 6).

$T_{CDW}$. Nevertheless, intense CDW reflections with propagation vector $\frac{1}{3}\frac{1}{3}\frac{1}{3}$ are observed below ~98 K, matching the x-ray data (Fig. 1I) and neutron scattering[41]. The reconstruction of the h h l plane is displayed in Fig. 2B at several temperatures. Besides confirming the absence of DS at high temperature at $l = \frac{1}{3}$, the $l = \frac{1}{2}$ precursor increases its intensity with l, indicating that the DS is driven by the condensation of an out-of-plane polarized mode. Dissecting the anisotropic diffuse signal with cuts along the l and h h directions around the $\mathbf{G}_{006}+(0\,0\,\frac{1}{2})$ position and fitting them to Lorentzian profiles, Fig. 2C, reveals a strong dependence of its intensity and linewidth with temperature. The correlation length of the diffuse precursor at $l = \frac{1}{2}$ diverges upon cooling and extends to nearly 20 Å in the $z$-direction and less than two unit cells in the $ab$-plane at low temperature. The critical exponent of the inverse correlation length along the $z$-direction, $\delta = 0.49 \pm 0.02$, follows the mean-field critical behavior characteristic of a second order phase transition. In the temperature range between 100 and 95 K, both the DS at $l = \frac{1}{2}$ and CDW satellites at $l = \frac{1}{3}$ coexist. With further cooling, the diffuse precursor vanishes and only the CDW peaks remain (Fig. 2E). On the other hand, DS experiments in YV₆Sn₆ did not reveal any diffuse signal, consistent with the absence of short-range charge precursors[5,51] (Supplementary Note 5).

In order to shed light on the microscopic origin of the DS at the $\frac{1}{3}\frac{1}{3}\frac{1}{2}$ reciprocal lattice vector, we have resolved the elastic and inelastic components of the scattered signal and studied the low energy lattice dynamics by means of inelastic x-ray scattering (IXS). Figure 3A compares the normalized IXS spectra with $\Delta E = 3$ meV (Supplementary Note 7, for fitting details) as a function of temperature, focusing on the $\frac{1}{3}\frac{1}{3}\frac{13}{2}$ region of the RS, where the DS develops its maximum intensity. At 300 K, the IXS spectrum consists on 2 low energy Stokes and anti-Stokes longitudinal modes with frequency ~3 meV, which correspond to the out-of-plane polarized Sn$^T$ atoms in trigonal coordination in the kagome lattice, according to our phonon calculations (Supplementary

Note 8). Their intensity grows upon cooling following the temperature behavior of the elastic central peak (CP, $E_{loss}$ ~ 0 meV) of IXS, Fig. 2D, thus the diffuse intensity contains spectral weight from the CP and phonons. Moreover, the generalized susceptibility, $\chi(q)$ (Fig. 2D inset), proportional to the **q** Fourier component of the displacement-displacement correlation function[52], follows a linear Curie-Weiss behavior for the quasi-elastic component of the DS and the low energy phonon, unlike the CP that deviates from the linear behavior at high temperature.

On the other hand, the phonon dispersion reveals an anomalous broad softening around $(h\,h) = (\frac{1}{3}\frac{1}{3})$ region of the momentum space, suggesting an incipient localization of the phonon fluctuations in real space, not captured by the phonon calculations, which shows a rather flat phonon band in the $k_z = \pi$ plane (and at other $k_z$'s). Similar **q**-dependence of the lattice dynamics has been observed in NbSe₂[31] and TiSe₂[32] at T > $T_{CDW}$ and also in high $T_c$ cuprates[53]. With further cooling, the momentum spread of the phonon softening acquires a U-shape, typical of systems which show an enhanced electron-phonon interaction (EPI), unlike the V-shape dispersion expected when the CDW is driven by the nesting of Fermi surface[25]. Nevertheless, although the low energy branch softens in a broad range of momenta, ~1 meV at the BZ center and ~2 meV at the BZ border, it only collapses at the critical wavevector $\mathbf{q} = \frac{1}{3}\frac{1}{3}\frac{1}{2}$ at $T_{CDW} = 98$ K. The temperature dependence of the frequency of the soft mode, plotted in Fig. 3D, returns a critical exponent of $0.47 \pm 0.07$, characteristic of the mean-field theory of phase transitions, and $T_c$ of 98 K, as observed experimentally. This demonstrates that the soft lattice dynamics of SVS showcase the first example of the weak coupling limit theory of a second-order phase transition within the kagome family. At the critical temperature, the IXS spectrum consists of a resolution-limited elastic line central peak at zero energy loss and the phonon is no longer resolvable (black scan at 98 K in 3A and Supplementary Note 7). On the other hand, the damping

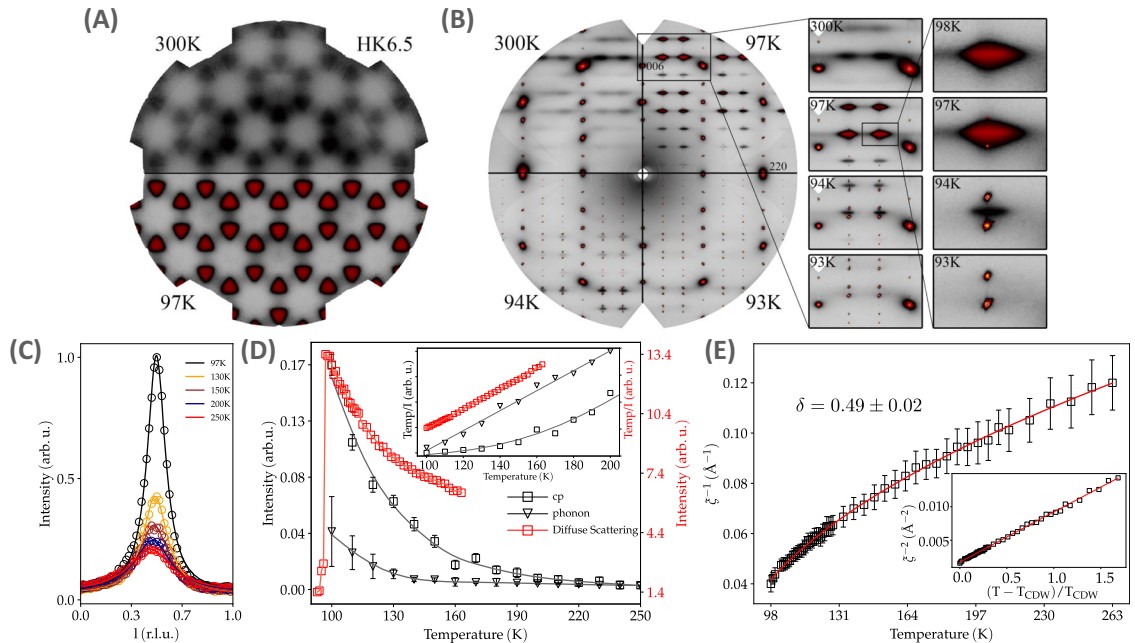

**Fig. 2 | Diffuse scattering maps of ScV₆Sn₆. A** h k 6.5 plane at 300 K (top) and 100 K (bottom), showing the diffuse precursor of the 3D CDW at $l = \frac{1}{2}$ that grows in intensity upon cooling. **B** h h l map, where no precursor is visible at $l = \frac{1}{3}$ at high temperature, but at $l = \frac{1}{2}$. The diffuse signal is replaced by the CDW Bragg satellites at low temperature, (right panels). **C** Temperature dependence of the diffuse scattering. Solid lines are fitted to Lorentzian profiles. **D** Temperature dependence of the diffuse intensity, IXS central peak (CP), and the low energy phonon at the $\frac{1}{3}\frac{1}{3}\frac{13}{2}$ position. Inset, the temperature dependence of the generalized susceptibility. **E** Temperature dependence of the inverse of the diffuse correlation length. Solid line is the fitting to the mean-field critical behavior, $\xi^{-1} = \xi_0^{-1}[(T - T_{CDW})/T_{CDW}]^\delta$ with $\delta = 0.49 \pm 0.02$. The error bars represent the fit uncertainty.

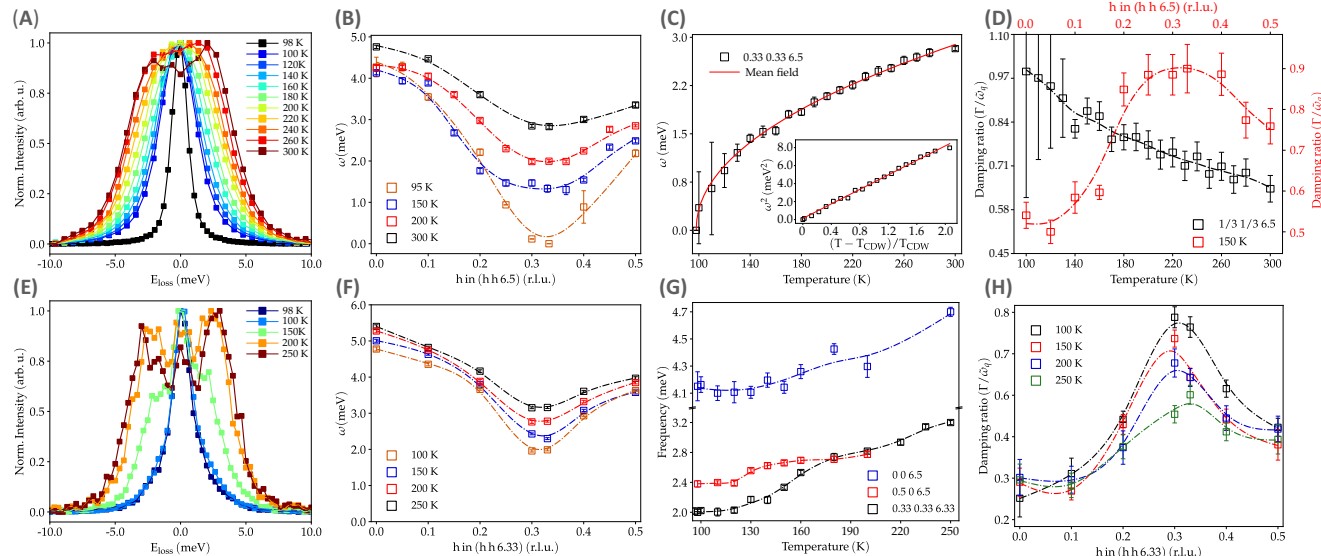

**Fig. 3 | Lattice dynamics of ScV₆Sn₆. A** Normalized temperature dependence of the IXS spectra at the $\frac{1}{3}\frac{1}{3}\frac{13}{2}$ r.l.u. position with energy resolution ΔE = 3 meV (ESRF). The normalized energy scan in black color corresponds to the IXS scan with an energy resolution of ΔE = 1.5 meV (APS). See Supplementary Note 7 for fitting details. **B** Momentum dependence of the $\frac{1}{3}\frac{1}{3}\frac{13}{2}$ phonon frequency at selected temperatures (ΔE = 3 meV), highlighting the large momentum softening. **C** Temperature dependence of the soft mode and its fitting to a power law with $\alpha = 0.47 \pm 0.07$. Inset, linear behavior of the squared frequency *vs* reduced temperature. **D** Temperature (momentum at 150 K) dependence of the damping ratio, $\Gamma/\bar{\omega}_q$, where $\Gamma$ is the damping and $\bar{\omega}_q$ the phonon frequency renormalized by the real part of the susceptibility (ΔE = 3 meV). **E** IXS scans as a function of temperature at the $\frac{1}{3}\frac{1}{3}\frac{19}{3}$ r.l.u. position (ΔE = 1.5 meV). **F** Momentum dependence of the $\frac{1}{3}\frac{1}{3}\frac{19}{3}$ phonon frequency as a function of temperature (ΔE = 1.5 meV). **G** Temperature dependence of the $\frac{1}{3}\frac{1}{3}\frac{19}{3}$ and 0 0 $\frac{13}{2}$ r.l.u. modes. Acoustic branches are shown in the Supplementary Note 7 (ΔE = 1.5 meV). **H** Momentum dependence of the damping ratio at selected temperatures. The error bars represent the fit uncertainty.

ratio of the soft mode increases upon cooling and becomes critically overdamped at the critical temperature (Fig. 3D), where phonons do not oscillate but only decay in time. Besides, the longitudinal low energy branch develops an anomalous broadening at high momentum

transfers (0.15 < h < 0.5 r.l.u.) with a damping ratio close to its critical value.

Having identified and characterized the dynamics of the soft mode, we now turn our attention to the lattice dynamics at the $\frac{1}{3}\frac{1}{3}\frac{1}{3}$

reciprocal lattice vector, where the electronic modulations develop at low temperature. In Fig. 3E, we present the temperature dependence of the normalized raw spectra at the $\frac{1}{3}\frac{1}{3}\frac{19}{3}$ lattice vector. The mode with an energy of 3 meV at $\mathbf{q}_{CDW}$ ($\Delta E = 1.5$ meV) is better resolvable and gradually softens throughout the entire region of momenta (~0.6 meV at the center and border of the BZ and ~1.2 meV at the $\mathbf{q}_{CDW}$) (Fig. 3F), but, without collapsing to zero frequency at $T_{CDW}$. Although the dispersion of the mode follows a more classic V-shape even at room temperature, its frequency remains finite at ~1.5 meV below 125 K and down to the critical temperature. Below $T_{CDW}$, the large enhancement of the elastic line, which drives the phase transition, completely masks the phonon and precludes its analysis. However, similar to the $l = \frac{1}{2}$ branch, the phonon again approaches the critical damping value at $\mathbf{q}_{CDW}$, demonstrating its coupling to electronic degrees of freedom. Indeed, the extraordinary coupling of electrons to the lattice is not only limited to the softening of the $l = \frac{1}{2}$ and $\frac{1}{3}$ branches, but also to the lattice vibrations at the A and L points of the BZ. The temperature dependence of the 0 0 6.5 and 0.5 0 6.5 modes, also involving the out-of-plane vibration of $Sn^T$ atoms, are displayed in Fig. 3G, showing a moderate softening from room temperature down to 125 K, thus demonstrating experimentally the richness of lattice instabilities explained by our calculations (Supplementary Note 7).

We now provide a theoretical understanding of the soft phonon modes. More and full details are found in Supplementary Note 8, 9, 10 and ref. 54. To begin with, in Fig. 4C, we show the high-temperature

phonon spectrum derived from our DFT calculation, which can also be understood as a non-interacting phonon spectrum where the renormalization from electron-phonon coupling almost vanishes. We observe the low-energy phonon branch, which will collapse at low temperatures, is extremely flat in most parts of the Brillouin zone and is formed mostly by the $z$-direction vibration of $Sn^T$ atoms (Fig. 4C). The flatness feature indicates the non-interacting phonon model can be described by a series of weakly coupled one-dimensional chains (as illustrated in Fig. 4A) formed by the $Sn^T$ atoms which are the heaviest atoms of the system and contribute most to the low-energy phonon mode. As we will show later, the picture of weakly coupled 1D chains still holds (with some acquired finite in-plane dispersion) even at low temperatures where significant renormalization from electrons appears.

Based on a recently introduced physical Gaussian approximation[55] and perturbation theory, we are able to analytically derive the electron correction to the non-interaction phonon spectrum[54] (Supplementary Note 9). The resulting phonon spectrum is shown in Fig. 4G with a comparison to the DFT-calculated phonon spectrum at zero temperature. We observe a good match, where both predict an imaginary phonon spectrum that is more dispersive (but still relatively flat) than the non-interacting phonon spectrum, with a leading order instability at experimentally observed phonon collapsing wave vector H = $\frac{1}{3}\frac{1}{3}\frac{1}{2}$.

We are able to analytically prove that the major correction to the non-interacting phonon spectrum originates from the coupling between the mirror-even phonon field $u_{ez}(\mathbf{R})$ and the density operator of mirror-even electron field $c_{\mathbf{R},e,\sigma}$, where $u_{ez}(\mathbf{R})$ and $c_{\mathbf{R},e,\sigma}$ are both even

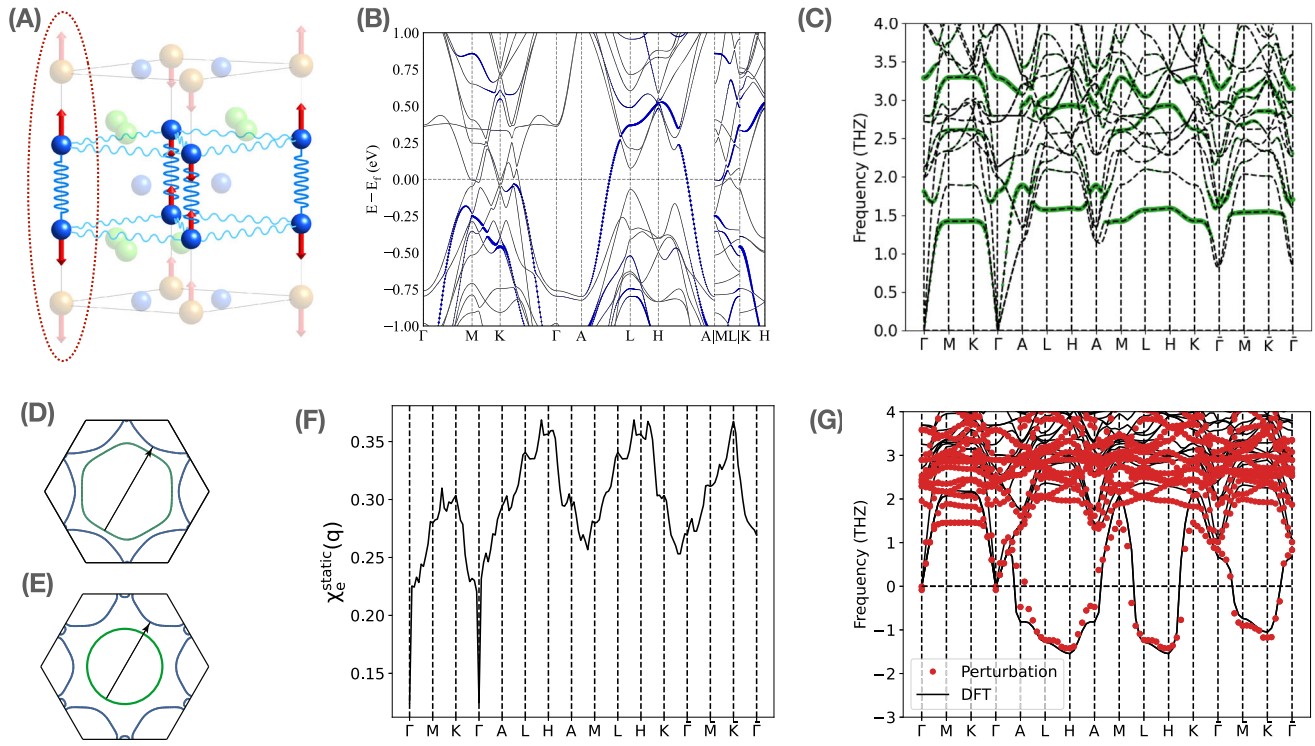

**Fig. 4 | Phonon spectrum, band structure, fermi surface nesting and charge susceptibility of ScV$_6$Sn$_6$. A** Vibration mode of the imaginary phonon mode at $\frac{1}{3}\frac{1}{3}\frac{1}{2}$. The phonon spectrum can be described by a series of weakly coupled 1D chains where the dashed red line marks one of the 1D chains. The $z$-direction vibration mode of the $Sn^T$ atom is only weakly coupled to the $z_g$-direction vibration modes of the other $Sn^T$ atom within the same plane since the displacement between two atoms is perpendicular to the direction of vibration. **B** Band structure of the pristine phase where the blue dots mark the weight of Sn $p_z$ orbitals. **C** High-temperature (non-interacting) phonon spectrum derived from DFT at T = 1.2 eV. The green dots mark the weights of the $z$-direction vibration of $Sn^T$ atoms, where we observe an

extremely flat low-energy phonon band that is mostly formed by the $z$-direction vibration of $Sn^T$ atoms. **D** Fermi surfaces at $k_z = -0.32$ (green) and $k_z = 0.18$ (blue) where we observe a weak nesting near H = $\frac{1}{3}\frac{1}{3}\frac{1}{2}$ (the exact nesting vector is (0.4, 0.4, 0.5)). **E** Fermi surfaces at $k_z = 0.20$ (green) and $k_z = 0.53$ (blue) where we observe a weak nesting with nesting vector $\bar{K} = \frac{1}{3}\frac{1}{3}\frac{1}{3}$. **F** Static charge susceptibility of the mirror-even electron operator $c_{\mathbf{R},e,\sigma}$. **G** Phonon spectrum derived from DFT at zero temperature and from perturbation theory by combining the non-interacting phonon spectrum with electron correction. Both methods predict a relatively flat imaginary mode at $k_z = \frac{1}{2}$ with leading-order instability at the wavevector $(\frac{1}{3}\frac{1}{3}\frac{1}{2})$ where we observe phonon collapsing experimentally.

under mirror-$z$ transformation and are defined as

$$u_{ez}(\mathbf{R}) = (u_{Sn_1^T,z}(\mathbf{R}) - u_{Sn_2^T,z}(\mathbf{R}))/\sqrt{2}$$
$$c_{\mathbf{R},e,\sigma} = (c_{\mathbf{R},(Sn_1^T,p_z),\sigma} - c_{\mathbf{R},(Sn_2^T,p_z),\sigma})/\sqrt{2} \quad (1)$$

where $u_{Sn_{1,2}^T,z}(\mathbf{R})$ denotes the phonon fields that characterize the $z$ direction movement of two $Sn^T$ atoms respectively, and $c_{\mathbf{R},(Sn_{1,2}^T,p_z),\sigma}$ denote the electron operator with $p_z$ orbital and spin $\sigma$ at two $Sn^T$ atoms respectively (see the Fig. 4B for the band structure and weight of $p_z$ orbital of $Sn^T$ atom). The $u_{ez}(\mathbf{R})$ field at unit-cell $\mathbf{R}$ represents the superposition of $z$-direction vibrations of two $Sn^T$ atoms within the same unit cell. The minus sign in the definition of $u_{ez}(\mathbf{R})$ in Eq. (1) marks the opposite moving direction of two $Sn^T$ atoms. Via the electron-phonon coupling, the charge fluctuations of mirror-even electron orbitals $c_{\mathbf{R},e,\sigma}$, which is characterized by the static charge susceptibility $\chi_e^{static}(\mathbf{q})$, will introduce a strong normalization to the $u_{ez}(\mathbf{R})$ phonon fields.

In Fig. 4F, we show the behavior of static charge susceptibility $\chi_e^{static}$, where we find two peaks, with one near H $= \frac{1}{3}\frac{1}{3}\frac{1}{2}$ point and the other one near $\bar{K} = \frac{1}{3}\frac{1}{3}3$ point. The enhancement of susceptibility near H and at $\bar{K}$ are attributed to the weak Fermi surface nesting as illustrated[56] in Fig. 4D, E. Even though we identify the peak structures in the $\chi_e^{static}(\mathbf{q})$, the peaks are weak and the overall susceptibility does not show a strong momentum dependency. Consequently, after transforming to the real space, the charge susceptibility can be approximately described by $\chi^e(\mathbf{R}) \approx \chi^{on\text{-}site}\delta_{\mathbf{R},0} + \sum_{i=1,\ldots,6}\chi^{xy}\delta_{\mathbf{R},\mathbf{R}_i^{xy}} + \sum_{i=1,2}\chi^z\delta_{\mathbf{R},\mathbf{R}_i^z}$, which consists of a strong on-site term $\chi^{on\text{-}site}$, a relatively weak nearest-neighbor in-plane term ($\chi^{xy}$) and also a weak nearest-neighbor out-of-plane term ($\chi^z$). $\mathbf{R}_i^{xy}$ and $\mathbf{R}_i^z$ characterize the in-plane and out-of-plane nearest-neighbor sites respectively. The on-site fluctuations $\chi^{on\text{-}site}$ lower the energy of $u_{ez}$ at all the momentum points, which is the main reason for the phonon collapsing. However, due to the inter-unit-cell coupling (along-$z$ direction) between $Sn^T$ phonon fields which come from both the non-interacting phonon model and the electron correction induced by $\chi^z$, the phonon spectrum realizes a minimum at $k_z = \frac{1}{2}$ plane and has a relatively flat imaginary phonon mode in the whole $k_z = \frac{1}{2}$ plane, which are mostly formed by the $u_{ez}$ phonon field. The relatively weak in-plane correlation $\chi^{xy}$ will only introduce a weak coupling between the 1D phonon chains and select the H points to be the leading-order instability. Finally, we mention that, at $\bar{K} = \frac{1}{3}\frac{1}{3}\frac{1}{3}$, $\chi^e$ also has a peak. However, the low-branch phonon bands at $k_z = \frac{1}{3}$ have less weight of $u_{ez}(\mathbf{R})$ compared to the low-branch phonon bands at $k_z = \frac{1}{2}$ due to the $z$-direction inter-unit cell coupling between the tridiagonal Sn phonon fields. Consequently, the low-branch phonon mode at $k_z = \frac{1}{3}$ experiences less renormalization from electron-phonon coupling and the dominant instability still locates at $k_z = \frac{1}{2}$ plane.

In summary, we have presented a comprehensive experimental and theoretical study of the electronic structure and lattice dynamics of the kagome $ScV_6Sn_6$. We have found, from diffuse scattering, a short-range CDW precursor with propagation vector $\frac{1}{3}\frac{1}{3}\frac{1}{2}$, indicating that the H point is the primary order parameter. A softening of a flat phonon plane at $k_z = \pi$, characterized by an out-of-plane vibration of the trigonal Sn atoms, drives the collapse of a soft mode at H and competes with the long-range CDW order at $\frac{1}{3}\frac{1}{3}\frac{1}{3}$. Moreover, we demonstrate that the low-energy phonon spectrum can be described by a series of weakly coupled 1D chains (at both high and—less so—at low temperatures), where the electron-phonon coupling and the charge fluctuation of the mirror-even orbital $c_{\mathbf{R},e,\sigma}$ drive the flat imaginary phonon modes at $k_z = \frac{1}{2}$ plane with the leading-order instability at the experimentally observed collapsing wavevector q $= \frac{1}{3}\frac{1}{3}\frac{1}{2}$, in agreement with the ab initio results and analytical effective models.

*Note added in proof*: During the review of this manuscript, an IXS study reported a softening, without collapse, of the $\frac{1}{3}\frac{1}{3}\frac{1}{2}$ low energy phonon mode in $ScV_6Sn_6$[57].

## Methods

High-quality single crystals of $ScV_6Sn_6$ were grown by flux method[41] using high purity starting elements. Sc, V and Sn were mixed in the molar ratio Sc:V:Sn of 1:6:60. We loaded all materials into an alumina crucible that was sealed inside a quartz tube under vacuum ($10^{-5}$ Torr). The tube was heated to 1100 °C over 10 h, soaked for 24 h, and then slowly cooled to 700 °C over 400 h. After centrifuging at 700 °C to remove excess Sn, the crystals were collected. Scanning electron microscopy with an energy-dispersive EDX analysis was used to determine the composition of the $ScV_6Sn_6$ crystal (Supplementary Note 2).

Diffuse scattering measurements (energy $E_i = 17.8$ keV) were performed at the ID28 beamline at the European Synchrotron Research Facility (ESRF) with a Dectris PILATUS3 1M X area detector. The CrysAlis software package was used for the orientation matrix refinement and reciprocal space reconstructions. Low energy phonons were measured by inelastic x-ray scattering (IXS) at the ID28 IXS station at the European Synchrotron Facility (ESRF), ($E_i = 17.8$ keV, $\Delta E = 3$ meV) and at the HERIX beamline at the Argonne Photon Source (APS) ($E_i = 23.72$ keV, $\Delta E = 1.5$ meV). The components ($hkl$) of the scattering vector are expressed in reciprocal lattice units (r.l.u.), $(h\,k\,l) = h\mathbf{a}^* + k\mathbf{b}^* + l\mathbf{c}^*$, where $\mathbf{a}^*$, $\mathbf{b}^*$, and $\mathbf{c}^*$ are the reciprocal lattice vectors.

ARPES experiments were performed at the ESM-ARPES beamline at the NSLS II synchrotron equipped with a Scienta DA30 electron energy analyzer. The samples were cleaved inside an ultra-high vacuum chamber with a base pressure better than $\sim 4 \times 10^{-10}$ torr and T = 7 K. The photon energy was scanned from 70 eV to 120 eV, from which we determine $\Gamma$ and A points of the BZ. The energy and momentum resolutions were better than 5 meV and 0.01 Å$^{-1}$.

Resonant hard x-ray resonant scattering experiments were performed at the beamline 4ID-D of the Advanced Photon Source at Argonne National Laboratory. High resolution x-ray diffraction was taken at the BM01 station of Swiss-Norwegian Beamlines at ESRF. The ab initio band structures are computed using the Vienna ab-initio Simulation Package (VASP)[58–62], where the generalized gradient approximation (GGA) of the Perdew-Burke-Ernzerhof (PBE)-type[63] exchange-correlation potential is used. A $8 \times 8 \times 6$ $\Gamma$-centered Monkhorst-Pack grid is adopted with a plane-wave energy cutoff of 400 eV. Wannier90[64–67] is used to construct maximally localized Wannier functions (MLWFs) for both the pristine and the CDW phases by considering the Sc $d$, V $d$, and Sn $p$ orbitals, and *WannierTools*[68] is used for post-processing.

## Data availability

Source data are provided with this paper. The scattering, ARPES, DS and IXS data generated in this study is available in the Figshare database: https://doi.org/10.6084/m9.figshare.24155937.

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

## Acknowledgements

We would like to thank Jiabin Yu and Jonah Herzog-Arbeitman for discussions. D.S. and S.B.-C. acknowledge financial support from the MINECO of Spain through the project PID2021-122609NB-C21 and by MCIN and by the European Union Next Generation EU/PRTR-C17.I1, as well as by IKUR Strategy under the collaboration agreement between Ikerbasque Foundation and DIPC on behalf of the Department of Education of the Basque Government. H.H. was supported by the European Research Council (ERC) under the European Union's Horizon 2020 research and innovation program (Grant Agreement No. 101020833). D.Căl. acknowledges the hospitality of the Donostia International Physics Center, at which this work was carried out. D.Că. and B.A.B. were supported by the European Research Council (ERC) under the European Union's Horizon 2020 research and innovation program (grant agreement no. 101020833) and by the Simons Investigator Grant No. 404513, the Gordon and Betty Moore Foundation through Grant No.GBMF8685 towards the Princeton theory program, the Gordon and Betty Moore Foundation's EPiQS Initiative (Grant No. GBMF11070), Office of Naval Research (ONR Grant No. N00014-20-1-2303), Global Collaborative Network Grant at Princeton University, BSF Israel US foundation No. 2018226, NSF-MERSEC (Grant No. MERSEC DMR 2011750). B.A.B. and C.F. are also part of the SuperC collaboration. C.F. acknowledges funding from the Deutsche Forschungsgemeinschaft (DFG, German Research Foundation) for 5249 (QUAST), the Deutsche Forschungsgemeinschaft (DFG) under SFB1143 (project no. 247310070) and the Würzburg-Dresden Cluster of Excellence on Complexity and Topology in Quantum Matter—ct.qmat (EXC 2147, project no. 390858490). This research used resources of the Advanced Photon Source, a U.S. Department of Energy (DOE) Office of Science user facility operated for the DOE Office of Science by Argonne National Laboratory under Contract No. DE-AC02-06CH11357. The research at the Electron Spectro Microscopy beamline (ESM-21ID) of the National Synchrotron Light Source II is supported by the U.S. Department of Energy (DOE) Office of Science User Facility, operated for the DOE Office of Science by Brookhaven National Laboratory under Contract No. DE-SC0012704.

## Author contributions

H.H., Y.J., D.Că., X.F., and B.A.B. developed the theoretical understanding of experimental observations. H.H., Y.J., and D.Că. performed analytical analysis of the model. Y.J., X.F., and M.G.V. performed ab initio calculations. C.Y., S.R., C.S., and C.F. synthesized the single crystals. A.K. and A.B. carried out the diffuse scattering and IXS measurements at ESRF, and A.H.S. and S.B.-C. at APS. S.B.-C. analyzed the IXS data. J.S., D.S., and S.B.-C. carried out the resonant x-ray scattering and A.B. and D.Ch. the high-resolution x-ray diffraction measurements and analyzed the data. A.R., E.V., and S.B.-C. conducted the ARPES measurements and D.S. analyzed and plotted the data. All authors wrote or provided input to the manuscript. B.A.B and S.B.-C. supervised and managed the project.

## Competing interests

The authors declare no competing interests.
