## [Peer Review File · Nature Communications]

REVIEWER COMMENTS

Reviewer #1 (Remarks to the Author):

By combination of the high-resolution elastic and inelastic x-ray (IXS) and diffuse scattering, angle-resolved photoemission spectroscopy (ARPES), density functional theory and analytical effective models, the authors unveiled the CDW instabilities in kagome metal ScV_6Sn_6 with low energy longitudinal phonon changing its wavevector around the CDW temperature without exhibiting long-range charge order. The broad low energy longitudinal phonon softening in momentum space indicates a strong momentum dependent electron-phonon coupling. They also showed that the underlying mechanism is related to the out-of-plane vibration of trigonal Sn and the longitudinal phonon mode starts flat at high temperature and gains dispersion due to the electron renormalization. The revealed dominated soft phonons and momentum dependence of CDW have not found in the intensively studied AV_3Sb_5 ($A = \text{K}, \text{Rb}, \text{Cs}$), thus providing a new platform for study intriguing CDW instability and soft phonon mode in kagome lattice.

This is an in-time work that definitely can stimulate much interest in further investigating CDW in ScV_6Sn_6 as well as interest in quantum manipulating the CDW to induce other intriguing emergent phenomena. The study is complete and rigorous owing to the combined strong technologies which give a very detailed and clear picture for the occurrence of CDW instabilities and the origin, which can warrant its publication in Nature Communications even in its current form. Nevertheless, before its acceptance, I would like to show below my several minor concerns:

1. The authors claimed that ScV_6Sn_6 possibly has four termination surfaces. For the ARPES measurements, which termination surface was measured?
2. The electronic band dispersion was measured in two samples and showed mismatch at the Γ point about 100 meV which is in fact very large. The authors ascribed it to polar surface termination in one of the two samples. This point is somewhat confusing. Why the termination surface is different? And which termination surface is not polar and the measurements gave reliable data?
3. The authors showed the results of measured electronic band structure. How about its comparison with other previously reported results?
4. The authors mentioned that they measured the EDAX data. How about the results? I can not find in the manuscript. Furthermore, it is naturally a question that how about the crystal quality? Have the authors done basic characterizations on the crystals used herein?

Reviewer #2 (Remarks to the Author):

Korshunov performed comprehensive study of a new charged ordered kagome metal ScV₆Sn₆. Symmetry breaking orders in kagome metals is a topical research direction in the hard condensed matter community. ScV₆Sn₆ is a newly discovered kagome metal that features charge density wave with a wavevector different from AV₃Sb₅ and FeGe families. While the absence of other symmetry-breaking orders, such as superconductivity and magnetism, potentially undermines its fundamental interests, ScV₆Sn₆ offered an opportunity to uncover the complex landscape of CDW mechanisms in various kagome metals.

In this work, the authors combined x-ray scattering techniques, angle-resolved photoemission spectroscopy and first principles calculations to understand the origin of CDW in ScV₆Sn₆. Their main experimental observation is that the large phonon softening at $(1/3, 1/3, 1/2)$ is different from the CDW wave-vector at $(1/3, 1/3, 1/3)$. Based on first principles calculations, the authors concluded that while the $(1/3, 1/3, 1/2)$ is the leading structural instability, the flatness of the unstable phonon energetically favors $(1/3, 1/3, 1/3)$ to suppress fluctuations through a first order phase transition.

Overall, I think it is a nice work. The experimental observations are interesting. The manuscript is potentially suitable for publication in Nature Communications if the following concerns are properly addressed.

1. Fig. 3a plots the normalized IXS curves as function of temperature. I find it is somewhat misleading as the elastic peak could be too strong to washout the weak inelastic signals. For instance, the higher resolution data show shoulders around 2 meV in Fig. 3E. The color gradient is also not very helpful to judge the experimental observation. Showing few representative curves with larger color contrast might be better.

2. Fig. 3D, I wonder how the error bars are determined. Near the CDW transition, the extracted phonon energy show larger uncertainty, it doesn't seem to be reflected in the damping ratio.

3. The IXS data were collected with different energy resolutions. It will be better to clarify which dataset is used for fitting.

4. It is still not clear to me why the CDW takes $(1/3, 1/3, 1/3)$ over $(1/3, 1/3, 1/2)$. The H-order can also induce first order phase transition. Could it be related to the stacking of CDW?

5. If I understand correctly, the "flat-phonon mode" is corresponding to the negative energy phonon in the calculation? If that's the case, I'm not sure if it is a properly way to describe it as in real materials, phonon modes have to be positive.

6. The title of the manuscript should be changed to "Softening of a flat phonon mode in the kagome ScV_6Sn_6 ".

7. page 6 line 4, $\text{textbf{q}}_{\text{CDW}}$ should be $\text{textbf{q}}_{\text{CDW}}$

Reviewer #1 (Remarks to the Author):

By combination of the high-resolution elastic and inelastic x-ray (IXS) and diffuse scattering, angle-resolved photoemission spectroscopy (ARPES), density functional theory and analytical effective models, the authors unveiled the CDW instabilities in kagome metal ScV₆Sn₆ with low energy longitudinal phonon changing its wavevector around the CDW temperature without exhibiting long-range charge order. The broad low energy longitudinal phonon softening in momentum space indicates a strong momentum dependent electron-phonon coupling. They also showed that the underlying mechanism is related to the out-of-plane vibration of trigonal Sn and the longitudinal phonon mode starts flat at high temperature and gains dispersion due to the electron renormalization. The revealed dominated soft phonons and momentum dependence of CDW have not found in the intensively studied AV₃Sb₅ (A = K, Rb, Cs), thus providing a new platform for study intriguing CDW instability and soft phonon mode in kagome lattice.

This is an in-time work that definitely can stimulate much interest in further investigating CDW in ScV₆Sn₆ as well as interest in quantum manipulating the CDW to induce other intriguing emergent phenomena. The study is complete and rigorous owing to the combined strong technologies which give a very detailed and clear picture for the occurrence of CDW instabilities and the origin, which can warrant its publication in Nature Communications even in its current form. Nevertheless, before its acceptance, I would like to show below my several minor concerns:

We thank the referee for his/her positive comments and his/her enthusiasm about our work. In the following, we will reply to his/her comments.

1. The authors claimed that ScV₆Sn₆ possibly has four termination surfaces. For the ARPES measurements, which termination surface was measured?

The $\Gamma \rightarrow M$ was measured with the ScSn₂ termination and the data along the $M \rightarrow K \rightarrow \Gamma \rightarrow M$ direction, with the V₃Sn kagome termination. This was determined by measuring the core level photoemission and comparing with the reports in the literature (Science Advances 8, eadd2024 (2022), <https://www.science.org/doi/pdf/10.1126/sciadv.add2024>). Details about the terminations are summarized in the supplementary information section 9, figure S22 of the SI and in the caption of figure 1.

2. The electronic band dispersion was measured in two samples and showed mismatch at the Γ point about 100 meV which is in fact very large. The authors ascribed it to polar surface termination in one of the two samples. This point is somewhat confusing. Why the termination surface is

different? And which termination surface is not polar and the measurements gave reliable data?

We thank the referee for pointing out this comment. The fact that the cleaving might end up in four different terminations, of which the V_3Sn and $ScSn$ are the most energetically favorable, does not explain the 80-100 meV offset at Gamma. Besides, the initial argument we gave in the manuscript, i.e polar surface does not hold either as none of the terminations are polar. We have been discussing with colleagues who have performed ARPES on this same material and conveyed us that they found offsets up to 50 meV, depending on the terminations and ascribed them to different doping levels during the growth process. Namely, the sample is grown by Sn flux, which is removed by centrifugation before cooling down, thus some Sn flux might still remain that results in areas with slightly different stoichiometry that is locally illuminated by the beam. The referee must note that the 80-100 meV offset is measured when the sample cleaves in the V_3Sn kagome plane ($M \rightarrow K \rightarrow \Gamma$ direction) and not when cleaving along the $ScSn_2$ plane, where DFT and ARPES fully match, thus it seems that the mismatch is surface termination dependent (these are crystals from the same batch).

We are completing the ARPES study, as we have detected signatures of electron-phonon interaction as kinks in the electronic dispersion (see figure below), which we are currently trying to quantify. Thus, the correlated nature of the ScV_6Sn_6 might also be responsible for the disagreement between DFT and the experimental data. Once we finish these experiments and their analysis, we will report these results in a separate manuscript reporting a comprehensive ARPES manuscript and its comparison with the available literature in ScV_6Sn_6 .

Figure: (A) Electronic band dispersion along the $\Gamma \rightarrow M$ direction and $k_2 = 6.1 \text{ \AA}^{-1}$, highlighting the kink feature (red) and the avoided band crossing (yellow). (B) Zoom in of the kink.

In any case, the mismatch between ARPES (surface sensitive technique) and DFT does not affect the phonon conclusions since the IXS and diffraction experiments are done in transmission geometry in single crystals of 200-micron thickness.

3. The authors showed the results of measured electronic band structure. How about its comparison with other previously reported results?

We have added a few lines in the main text comparing our ARPES with the results published in the literature and updated the bibliography with the recent ARPES results from other groups (works carried out after the submission of this manuscript). Nonetheless, as stated above, there will be a second manuscript reporting a comprehensive electronic characterization of our samples.

4. The authors mentioned that they measured the EDAX data. How about the results? I cannot find in the manuscript. Furthermore, it is naturally a question that how about the crystal quality? Have the authors done basic characterizations on the crystals used herein?

Yes, we have performed EDX, powder x-ray diffraction and transport characterization. The X-ray and EDAX data are now included in the supplementary information. The transport data of the crystals is shown in the manuscript [arXiv:2305.04683](https://arxiv.org/abs/2305.04683)

Reviewer #2 (Remarks to the Author):

Korshunov performed comprehensive study of a new charged ordered kagome metal ScV_6Sn_6 . Symmetry breaking orders in kagome metals is a topical research direction in the hard condensed matter community. ScV_6Sn_6 is a newly discovered kagome metal that features charge density wave with a wavevector different from AV_3Sb_5 and FeGe families. While the absence of other symmetry-breaking orders, such as superconductivity and magnetism, potentially undermines its fundamental interests, ScV_6Sn_6 offered an opportunity to uncover the complex landscape of CDW mechanisms in various kagome metals.

In this work, the authors combined x-ray scattering techniques, angle-resolved photoemission spectroscopy and first principles calculations to understand the origin of CDW in ScV_6Sn_6 . Their main experimental observation is that the large phonon softening at $(1/3, 1/3, 1/2)$ is different from the CDW wave-vector at $(1/3, 1/3, 1/3)$. Based on first principles calculations, the authors concluded that while the $(1/3, 1/3, 1/2)$ is the leading structural instability, the flatness of the unstable phonon energetically favors $(1/3, 1/3, 1/3)$ to suppress fluctuations through a first order phase transition.

Overall, I think it is a nice work. The experimental observations are

interesting. The manuscript is potentially suitable for publication in Nature Communications if the following concerns are properly addressed.

We acknowledge the referee for his/her positive evaluation of our work. Next, we will reply his/her comments.

1. Fig. 3a plots the normalized IXS curves as function of temperature. I found it is somewhat misleading as the elastic peak could be too strong to washout the weak inelastic signals. For instance, the higher resolution data show shoulders around 2 meV in Fig. 3E. The color gradient is also not very helpful to judge the experimental observation. Showing few representative curves with larger color contrast might be better.

We thank the referee for his/her comment. At low temperature, the elastic line is strong but still we can resolve phonons. In the supplementary information figure S6-S10, we plot and compared the unrenormalized data and details of the analysis (see for instance figure S7(A)). Although the data at $1/3, 1/3, 1/2$ was measured with 3 meV resolution (ESRF), we have taken representative scans at APS ($\Delta E=1.5$ meV, figure S7(B)) that confirm the absence of phonon at T_{CDW} , within the experimental resolution of $\Delta E=1.5$ meV, despite the large enhancement of the elastic line. Moreover, the softening of the phonon follows a mean field behavior (consistent with the divergence of the correlation length measured in DS), thus it is expected a full collapse of the low energy mode at T_{CDW} . Due to the softening of the phonon and the resolution of the experiment, the collapse of the mode at $T \sim T_{CDW}$ is always washed out by the enhancement of the elastic line at T_{CDW} . We have changed the color code of figures 3A and E of the main text for better visualization.

2. Fig. 3D, I wonder how the error bars are determined. Near the CDW transition, the extracted phonon energy show larger uncertainty, it doesn't seem to be reflected in the damping ratio.

Error bars are the result of the fitting uncertainty of the frequency and linewidths. On the other hand, the data in figure 3D did not contain the error bar. These have now been added.

3. The IXS data were collected with different energy resolutions. It will be better to clarify which dataset is used for fitting

We have now clarified in the caption of the figures and in the main text which data has been measured with each energy resolution.

4. It is still not clear to me why the CDW takes $(1/3, 1/3, 1/3)$ over $(1/3, 1/3, 1/2)$. The H-order can also induce first order phase transition. Could it be related to the stacking of CDW?

We thank the referee for giving us the opportunity to further clarify this point. Among all the possible competing CDW wave vectors, $(1/3, 1/3, 1/3)$ is the only wave vector at which the order parameter can have a cubic term in the Landau Ginzburg free energy, hence inducing a first-order transition

at this wave vector. Thus, $(1/3, 1/3, 1/3)$ will be a natural choice of CDW, since the fluctuation induced by the flat phonon band can be suppressed by the first-order transition.

From symmetry, the order parameter with wave vector $H=(1/3, 1/3, 1/2)$ cannot have a cubic term in the Landau Ginzburg free energy. Therefore, it is unlikely that H-order can induce a first-order transition due to the absence of a cubic term.

A more detailed explanation and a theoretical calculation of the CDW transition are given in a theoretical paper arXiv:2305:15469.

5. If I understand correctly, the "flat-phonon mode" is corresponding to the negative energy phonon in the calculation? If that's the case, I'm not sure if it is a properly way to describe it as in real materials, phonon modes have to be positive.

The flat phonon mode is negative in the calculation, but positive in the real material. The negative energy squared in the calculation indicates the possible instabilities of the system at zero temperature. As we increase the temperature, the energy squared of the phonon increases and become positive at high temperature. Moreover, the theoretical calculations at zero temperature (note that the temperature included in the calculations is electronic temperature that does not represent a real temperature scale) could help us to understand why the phonon softens and collapses in the experiments. Here, we observe a flat phonon band near the H point gives the leading-order instability in the phonon calculation, which is consistent with the experiment. And our theoretical calculation tells us the softening of the flat phonon band comes from the electron-phonon coupling between mirror-even electron orbitals and mirror-even phonon modes. If one does the proper calculation, for instance considering the Stochastic Self Consistent Anharmonicity (SSCHA theory) one would obtain positive flat phonon modes for $T > T_{CDW}$.

6. The title of the manuscript should be changed to "Softening of a flat phonon mode in the kagome $ScV_{1-x}Sn_x$ ".

We have modified it.

7. page 6 line 4, q_{CDW} should be q_{CDW}

We have corrected the typo.

REVIEWERS' COMMENTS

Reviewer #1 (Remarks to the Author):

I am satisfied with the responses and revisions made by the authors. The work provides a comprehensive physical picture for the CDW of ScV_6Sn_6 . I would like to recommend the publication in NC.

Reviewer #2 (Remarks to the Author):

In the revised manuscript, the authors properly addressed my comments and suggestions. The current paper provides new and important insight on the diverse CDW mechanisms in Kagome metals. I, therefore, happy to recommend its publication in Nature Communications.